# Susceptibility and Virulence of *Enterobacteriaceae* Isolated from Urinary Tract Infections in Benin

**DOI:** 10.3390/microorganisms11010213

**Published:** 2023-01-14

**Authors:** Funkè F. Assouma, Haziz Sina, Tomabu Adjobimey, Agossou Damien Pacôme Noumavo, Akim Socohou, Bawa Boya, Ange D. Dossou, Lauriane Akpovo, Basile Boni Saka Konmy, Jacques F. Mavoungou, Adolphe Adjanohoun, Lamine Baba-Moussa

**Affiliations:** 1Laboratory of Biochemistry and Molecular Typing in Microbiology, Department of Biochemistry and Cell Biology, Faculty of Science and Technology, University of Abomey-Calavi, Abomey-Calavi 05 BP 1604, Benin; 2Laboratory of Biochemistry and Molecular Biology, Department of Biochemistry and Cell Biology, Faculty of Science and Technology, University of Abomey-Calavi, Abomey-Calavi 05 BP 1604, Benin; 3Laboratory of Microbiology and Food Technologies, Department of Plant Biology, Faculty of Science and Technology, University of Abomey-Calavi, Cotonou 04 BP 1107, Benin; 4Benin Ministry of Heath, Cotonou 01 BP 363, Benin; 5Department of Microbiology, International University of Libreville, ESSASSA-Libreville Campus, Essassa BP 20411, Gabon; 6National Agronomic Research Institute of Benin, Cotonou 01 BP 884, Benin

**Keywords:** urinary tract infections, *Enterobacteriaceae*, resistance, biofilm, ESBL, virulence, Benin

## Abstract

*Enterobacteriaceae* represent one of the main families of Gram-negative bacilli responsible for serious urinary tract infections (UTIs). The present study aimed to define the resistance profile and the virulence of *Enterobacteriaceae* strains isolated in urinary tract infections in Benin. A total of 390 urine samples were collected from patients with UTIs, and *Enterobacteriaceae* strains were isolated according to standard microbiology methods. The API 20E gallery was used for biochemical identification. All the isolated strains were subjected to antimicrobial susceptibility testing using the disc diffusion method. Extended-spectrum beta-lactamase (ESBL) production was investigated using a double-disc synergy test (DDST), and biofilm production was quantified using the microplate method. Multiplex PCR was used to detect uro-virulence genes, namely: *Pap*G, *Iron*B, *Sfa*, *iuc*D, *Hly*, *Foc*G, *Sat*, *Fyu*A and *Cnf*, using commercially designed primers. More than 26% (103/390) of our samples were contaminated by *Enterobacteriaceae* strains at different levels. Thus, *E. coli* (31.07%, 32/103), *Serratia marcescens* (11.65%, 12/103), *Klebsiella ornithinolytica* (8.74%, 9/103), *Serratia fonticola* (7.77%, 8/103) and *Enterobacter cloacae* (6.80%, 7/103) were identified. Among the isolated strains, 39.81% (41/103) were biofilm-forming, while 5.83% (6/103) were ESBL-producing. Isolates were most resistant to erythromycin, cefixime, ceftriaxone and ampicillin (≥90%) followed by ciprofloxacin, gentamycin, doxycycline and levofloxacin (≥50%), and least resistant to imipenem (27.18%). In regard to virulence genes, *Sfa* was the most detected (28.15%), followed by *IronB* (22.23%), *iucD* (21.36%), *Cnf* (15.53%), *PapG* (9.71%), *FocG* (8.74%), *Sat* (6.79%), *FyuA* (5.82%) and *Hyl* (2.91%). These data may help improve the diagnosis of uropathogenic strains of *Enterobacteriaceae*, but also in designing effective strategies and measures for the prevention and management of severe, recurrent, or complicated urinary tract infections in Benin.

## 1. Introduction

Urinary tract infections (UTIs) affect nearly 250 million people yearly and represent approximately 40% of infections worldwide. They account for 10–20% of nosocomial infections [1,2]. UTIs are associated with considerable morbidity and a large spectrum of clinical symptoms, ranging from asymptomatic bacteriuria to cystitis or septic shock, that can lead to life-threatening multiple-organ failure [3]. Most UTIs have a bacterial origin, and the most frequent cause of the infection is *Enterobacteriaceae* [4]. The most commonly encountered *Enterobacteriaceae* are *Escherichia coli*, *Klebsiella pneumoniae* and *Enterobacter cloacae* [5].

The ability of *Enterobacteriaceae* to invade and persist in the uroepithelium depends on several virulence factors and their ability to form biofilms [6]. Biofilm-forming bacteria are a common cause of recurrent and severe urinary tract infections and are generally multidrug-resistant bacteria [7]. In addition to the formation of biofilms, resistance to empirical antimicrobial treatments has increased in recent years [8], especially in Gram-negative bacteria [9]. Several studies have shown an increase in antimicrobial resistance to the most commonly used antibiotics, such as ciprofloxacin and trimethoprim-sulfamethoxazole, in strains of *Enterobacteriaceae* isolated from urinary tract infections [10,11]. Recent studies in Africa and Europe reported a substantial increase in Gram-negative bacteria from ESBL-producing urinary tract infections [12,13]. Indeed, the spread of ESBL-producing bacteria has made the empirical treatment of infections more difficult and has promoted resistance to beta-lactam antibiotics such as penicillin, cephalosporins and sometimes even carbapenems [14].

The pathogenicity of *Enterobacteriaceae* in urinary tract infections increases with the presence of virulence factors. Indeed, *Enterobacteriaceae* strains harbor several virulence genes associated with serious or recurrent urinary tract infections [15]. Among these genes, P fimbriae (*pap*), S-fimbriae (*sfa*), hemolysins (*hly*), cytotoxic-necrotizing-factor (*cnf1*) and Aerobactin (*iucD*) are the most relevant [16]. While *pap* and *sfa* genes are well known to promote docking, factors associated with the colonization of the host [17], the *hly*, *cnf1* and *fyuA* genes, are mainly associated with intracellular survival, iron acquisition, immune system leakage, the inflammatory response and host tissue damage [18,19].

Effective management and treatment of urinary tract infections require an in-depth understanding of antimicrobial resistance, virulence genes and biofilm formation in strains of *Enterobacteriaceae* isolated from urinary tract infections [20,21]. Understanding the link between biofilm formation, the presence of virulence genes and the distribution of antimicrobial resistance in strains of *Enterobacteriaceae* implicated in urinary tract infections will also allow for more effective prevention and management strategies [20]. Thus, the present study analyzed resistance profiles and virulence factors associated with *Enterobacteriaceae-related* urinary tract infections in Benin.

## 2. Materials and Methods

### 2.1. Urine Sample Collection

The sample size was determined using the Schwartz [22] formula $n=\frac{t^{2}\times \;p\left(1-p\right)}{m^{2}}$, with n = required sample size, t = 95% confidence level (typical value of 1.96), p = the prevalence of urinary tract infections (11.7%) and m = 5%. The urine samples (390) used in this study included samples from hospitalized patients and outpatients with clinical symptoms suggestive of a possible urinary tract infection and were obtained before the start of any antimicrobial treatment. Samples from patients under antibiotic therapy were not taken into account in our study. Sample collection was performed between March 2021 and March 2022 in 9 hospitals, namely the Natitingou Area Hospital (n = 75), Djougou Area Hospital (n = 10), Ménontin Area Hospital (n = 30), Departmental Hospital and University Centers of Borgou/Alibori (n = 90), Departmental Hospital Centers of Zou/Colline (n = 50), Departmental Hospital Centers of Mono/Couffo (n = 70), Departmental Hospital and University Centers of Ouémé/Plateau (n = 25), Clinique bon Samaritan of Porto-Novo (20) and Clinique Senalia (n = 20). 

### 2.2. Isolation and Identification of Enterobacteriaceae Strains

Once collected, urine samples were cultured on different media, including blood agar, Eosin Methylene Blue (EMB) agar and nutrient agar, and incubated at 37 °C for 24 h. After the incubation time, suspected colonies were stained using the Gram staining method. In addition, the shapes, colors and arrangements of the colonies were observed [23]. The identification of *Enterobacteriaceae* species was performed using 23 biochemical tests (0- nitrophenyl-fi-D-galactosidase, arginine di-hydrolase, lysine and ornithine decarboxylase, citrate utilization, hydrogen sulfide, urease, tryptophan deaminase, indole, Voges–Proskauer, gelatin liquefaction, fermentation of glucose, mannitol, inositol, sorbitol, rhamnose, sucrose, melibiose, amygdalin and arabinose, nitrate reduction and nitrogen gas production, and catalase production) on an API 20E (BioMerieux SA, Marcy-l’Etoile, France) strip. 

### 2.3. Antibiotic Susceptibility of Isolates

The susceptibility of isolated *Enterobacteriaceae* to 12 antibiotics was tested using the disc diffusion method on Mueller Hinton agar medium, in accordance with the recommendations of the Antibiogram Committee of the French Society of Microbiology [24]. The bacterial suspension was standardized using the McFarland 0.5 control. The antibiotics studied were ampicillin (AMP, 10 μg), cefuroxime (CXM, 30 μg), amoxicillin + clavulanic acid (AMC, 30 μg), gentamicin (G, 10 μg), ciprofloxacin (CIP, 5 μg), ceftriaxone (CRO, 30 μg), cefixime (CFM, 5μg), levofloxacin (LEV, 5 μg), sulfonamide (SSS, 300 μg), erythromycin (E, 15 μg), imipenem (IPM, 10 μg) and doxycycline (DO, 30 μg). 

### 2.4. ESBL Production Detection Tests

The extended-spectrum beta-lactamase (ESBL) production test was carried out with 3rd generation cephalosporins, namely cefuroxime (CXM) and ceftriaxone (CRO) in the presence of amoxicillin + clavulanic acid (AMC) placed in the center of two cephalosporin discs. The result was considered positive if potentiation of the corkscrew-shaped zone of inhibition between the discs of CXM and AMC, and that of AMC and CRO, was observed [25].

### 2.5. Detection of the Bacterial Ability to Form Biofilm

The in vitro ability of isolated *Enterobacteriaceae* to form biofilm was determined using the method previously described by Christensen et al. [26]. Briefly, a 48-well microplate was inoculated with 10 μL of 18 h bacterial suspension to which 150 μL of Brain–Heart Infusion (BHI) was added. The microplates were incubated for 24 h at 37 °C, and then, the wells were washed three times with 0.2 mL of sterile physiological water in order to eliminate free bacteria. The biofilms formed by the adhesion of the sessile organisms to the microplate wells were stained with crystal violet (0.1%) for 10 min [27]. After drying in the open air, the appearance of a visible film on the walls of the microplates and the bottom of the walls was considered an indication of biofilm production.

### 2.6. Molecular Identification of Virulence Factors

DNA extraction was performed using the method of Rasmussen and Morrissey [28] and multiplex PCR was used for virulence gene detection. The amplification was carried in a 20 µL mix containing 2.0 µL of buffer (10×), 0.4 µL of MgCl_2_ (25 mM), 0.2 µL of dNTPs (10 mM), 1 µL of primer F (10 mM), 1 µL of primer R (10 mM), 0.2 µL of *Taq* DNA polymerase, and DNA (5 µL) under initial denaturation conditions of 94 °C for 5 min, followed by 30 cycles of denaturation (60 s at 94 °C), annealing (60 s at 53 °C) and elongations (60 s at 72 °C), followed by final elongation at 72 °C for 10 min. After amplification, PCR products were migrated on 1.2% agarose gel for about 30 min at 100 V. A 100 bp molecular weight marker (Gene Ruler) was used. The primers sequences [29] used to target the detection of virulence factors are shown in Table 1. 

### 2.7. Data Analysis

Means and standard deviations were calculated from the experimental results using an Excel 2013 spreadsheet. Graph Pad Prism 8 software was used to determine significant differences at the 5% threshold (*p* ˂ 0.05) between the calculated means. In addition, resistant and biofilm-forming species possessing virulence genes were subjected to principal component analysis (PCA) using R 4.2.1 software to determine the correlation between resistant, biofilm-forming and virulence genes.

## 3. Results

### 3.1. Sociodemographic Characteristics of Patients

The age of the majority of the patients (22.23%) included in the study ranges from 21 to 30 years (Table 2). The average age was 41 years old and the maximum was 76 years old. Positive samples for *Enterobacteriaceae* were mainly seen in female patients (64.08%). The sex ratio of M/F was 0.56. 

### 3.2. Enterobacteriaceae Strains’ Isolation Frequency

Out of the 390 urine samples collected, 103 (26.41%) were contaminated with *Enterobacteriaceae* strains. Remarkable diversity was observed among these *Enterobacteriaceae* strains (Figure 1). A total of 18 species of *Enterobacteriaceae* were identified with predominance of *Escherichia coli* (32/103, 31.07%), followed by *Serratia marcescens* (12/103, 11.65%) and *Klebsiella ornithinolytica* (9/103, 8.74%). The less represented species were *Serratia liquefaciens*, *Citrobacter freundii*, *Enterobacter intermidius* and *Klebsiella pneumoniae* (1/103, 0.97%). There was non-significant variation (*p* < 0.05) in the different species according to sex.

### 3.3. Antibiotic Susceptibility of Enterobacteriaceae Strains

The antibiotic resistance of *Enterobacteriaceae* isolates is shown in Figure 2. The results showed variable (between 27.18% and 100%) susceptibility of isolated *Enterobacteriaceae* strains to the tested antibiotics. Indeed, the highest strain resistance rates were observed with antibiotics such as erythromycin (100%), cefixime (99.02%), ampicillin (96.11%) and amoxicillin/clavulanic acid (88.35%). However, a low resistance rate was observed with imipenem (27.18%). In addition, the results indicated that all *Enterobacter sakazakii* strains are resistant to gentamicin, ciprofloxacin, augmentin, cefuroxime, erythromycin, ceftriaxone, levofloxacin, sulfonamide, doxycycline and cefixime. All the isolated *Escherichia coli* were resistant to erythromycin, ampicillin, ceftriaxone and cefixime. In contrast, all *Citrobacter freundii* and *Enterobacter intermidius* were sensitive to gentamicin, doxycycline and imipenem (Table 3). The analysis of variance showed that there was a significant difference between the resistance of species and antibiotics (*p* < 0.0001). 

### 3.4. ESBL Production by Isolated Enterobacteriaceae Strains

Of the 103 bacterial isolates, 6% were ESBL producers (Figure 3).

### 3.5. Biofilm Formation Capability

The biofilm formation test revealed that 40% of the strains were biofilm-forming. In regard to the species, isolated *Enterobacter clocae* strains were the most efficient at biofilm formation (57%) followed *by Serratia odolifera* (50%), *Enterobacter gergoviae* (50%), *Salmonella arizonae* (50%), *Enterobacter sakazakii* (50%), *Salmonella* spp. (50%) and *Escherichia coli* (48.38%) (Figure 4). 

### 3.6. Detection of Virulence Genes

Virulence genes were detected in 64.07% of *Enterobacteriaceae* isolates. Thus, genes encoding for S-frimbria adhesin (28.15%), salmochelin outer membrane receptor (22.23%), aerobactin biosynthesis receptor (21.36%), cytotoxic necrotizing factor type 1 (15.53%), adhesin P-fimbria (*PapG*) (9.71%), F1C fimbrial adhesin (8.74%), secreted autotransporter toxin (6.79%), yersiniabactin outer membrane receptor (5.82%) and hemolysin RTX (2.91%) were found in various proportions (Figure 5). 

Considering the presence of nine targeted virulence genes by species, *Escherichia coli* isolates harbored eight, namely *Sfas* (25.8%), *IronB* (32.25%), *iucD* (35.48%), *Cnfl* (19.35%), *PapG* (9.67%), *FocG* (16.12%), *Sat* (6.45%) and *FyuA* (12.9%), in different proportions. Among *Serratia marcescens* isolates, six virulence genes (*Sfas*: 41.66%, *IronB*: 25%, *iucD*: 16.66%, *Cnfl*: 33.33%, *PapG*: 25% and *Sat*: 16.66%) were reported to be present. Only one gene was found among the isolates of *Citrobacter freundii*, *Enterobacter gergoviae* and *Serratia odolifera* (Table 4). The analysis of variance showed a significant difference between the presence of the virulence factors of a species (*p* < 0.0173) and a highly significant difference between the species and the virulence genes (*p* < 0.0039).

### 3.7. Relationship between Virulence Genes and Antibiotic Resistance

The analysis of the eigenvalues of the correlation matrix reveals that the first two components explain 69.23% of the variability (Table 5). Since this share of information is greater than 50%, the first two components can be used to adequately interpret the results of the Principal Component Analysis (PCA). 

Correlation analyses between the two components and the initial variables (Table 6 and Figure 6) shows at the level of axis 1, a strong positive correlation with the variables “*IRON* B”, “*SAT*” and “*FocG*” and a strong negative correlation with the variables “LEV”, “CN”, “CIP” and “SSS” are positively correlated with axis 1. Thus, axis 1 expresses that the resistance of the isolated strains to the antibiotics “LEV”, “CN “, “CIP” and “SSS” is linked to the absence of the virulence genes “*IRON* B”, “*SAT*” and “*FocG*”. As for axis 2, it shows a positive correlation with the “AMC” and “DO” variables and a negative correlation with the “Pap G” variable. This axis indicates that the presence of the “pap G” virulence gene leads to the sensitivity of the “AMC” and “DO” strains to antibiotics.

The projection of the different observations at the level of axes 1 and 2 (Figure 7) indicates that the species *E. cloacae*, *E. sakazakii*, *K. ornithinolytica*, and *K. terrigena* are mainly located in the positive parts of the two axes. These species are resistant to the antibiotics “DO”, “CN” and “SSS”. *C. freundii* and *S. liquefaciens* are mainly located in the positive part of axis 1 and the negative part of axis 2. *S. liquefaciens* better expresses the virulence gene Pap G. *E. intermidius* is located in the negative part of the two axes. This species expresses *sat* and *Forc* G virulence genes, which were not found in *E. cloacae*, *E. sakazakii*, *K. ornithinolytica* and *K. terrigena*. The species *S. arizonae* and *S. fonticola* are located in the negative part of axis 1 and the positive part of axis 2. It was noticeable that these species express both the *Iron*B virulence gene and the AMC resistance gene, in contrast to the species *C. freundii* and *S. liquefaciens*.

### 3.8. The Relationship between Biofilm Production and Virulence Genes

Correlation analyses between biofilm production, virulence genes and resistance showed a positive correlation between biofilm production and the virulence gene *Fyu*A and resistance to gentamicin. The more a species express *Fyu*A (virulence gene and resistant to gentamicin), the more biofilm it produces. A positive correlation was also found between biofilm production and sex. Species isolated from male UTIs produced more biofilm than species isolated from female UTIs. The virulence genes *Pap*G, *iuc*D and *Iron*B are negatively correlated with biofilm production. Therefore, the presence of the virulence genes *Pap*G, *iuc*D and *Iron*B decreases the production of biofilm.

## 4. Discussion

To better understand the pathogenicity of strains causing infections and develop new vaccines and therapeutic targets, it is necessary to identify the susceptibility to antibiotics, the factors associated with the formation of biofilms and the virulence factors of these strains [7,30]. These potential predictors help clinicians manage patients and anticipate the evolution of the infection in the host organization [31]. In this study, we sought to determine the prevalence of *Enterobacteriaceae* strains, their antibiotic resistance profiles, their ability to form biofilms and the presence of some urinary tract infection virulence genes in Benin.

During our study, *Enterobacteriaceae* were isolated from patients mainly between 21 and 30 years old (22.23%). This age bias can be explained by the fact professional, cultural and sporting activities may contribute to the infection. The results of our study indicate the presence of *Enterobacteriaceae* strains in a proportion of 26.41% in urine samples. Our results are lower than those obtained in Mali [32], which found a prevalence of 76.7% of *Enterobacteriaceae* in urine. *Enterobacteriaceae* have also been shown to cause 84–87% of UTIs [33]. This difference may be due to sample size, laboratory strain detection techniques, social demographics, climatic conditions, people’s levels of personal hygiene and healthcare-seeking habits. *Enterobacteriaceae* are found in the urine because they can easily contaminate the urinary tract, especially in women, since they are normal flora of the large intestine [7].

In our study, 18 different species of *Enterobacteriaceae* were isolated, with a predominance of *Escherichia coli* (30.97%), followed by *Serratia marcescens* (11.65%) and *Klebsiella ornithinolytica* (8.73%). On the other hand, the weakly represented species were *Serratia liquefaciens*, *Citrobacter freundii* and *Enterobacter intermidius* (0.97%). As expected, *E. coli* was the major *Enterobacterales* species among the urinary tract samples, which is similar to previous works in other countries [34,35]. However, the rate of 30.97% in this study is lower than the 72% recorded in France [36]. Traditionally, *E. coli* has been the dominant uropathogen due to its expression of toxins, adhesins, pili, and fimbriae that allow it to adhere to the uroepithelium [37]. These protect bacteria from urinary elimination and allow for bacterial multiplication and invasion of uroepithelial tissues. The presence of *S. marcescens* as a second isolated species can be explained by the fact that it has a great affinity for the urinary tract [38].

The resistance of *Enterobacteriaceae* isolates varied according to the antibiotics. We observed variable resistance rates depending on the families of antibiotics and the species isolated. High resistance rates (≥80%) were obtained with molecules of the macrolide, cephalosporin and penicillin families. Resistance rates of more than 50% were obtained with molecules from the fluoroquinolone, aminoglycoside and tetracycline families. In a recent study conducted in Benin on surgical patients, a high resistance rate to many of the antibiotics tested was shown [39]. However, our results are superior to those found in Senegal [40] and close to those obtained in Algeria [41]. Antibiotic resistance and the rapid spread of aminoglycosides and lactams such as cephalosporins and fluoroquinolones against uropathogenic bacteria compromise the clinical management of the infection and lead to a poor prognosis [42]. The multiple-drug resistance (MDR) of pathogenic bacteria may be associated with severe morbidity in urinary tract infections, leading to a major global health problem [43,44,45]. The non-regulation of the use of antibiotics in patients with access to over-the-counter prescriptions, the misuse of certain classes of antimicrobials, frequent self-medication with often random and inappropriate dosages, the premature discontinuation of treatment, the use of antibiotics as growth promoters in agriculture, the use of contraband molecules that are often less dosed or devoid of active ingredients, and unfavorable economic and social conditions are the main drivers that promote the emergence of bacteria that are multi-resistant to antibiotics [46,47]. In order to control the spread of this antibiotic resistance, since it represents a serious health problem in Benin, actions to raise awareness of the proper use of antibiotics coupled with monitoring of the acquisition of antibiotics must be implemented. Weak resistance to imipenem was detected. This trend was also found by Hashemi et al. [48], with an *Enterobacteriaceae* resistance rate to imipenem of around 19%. Carbapenems, therefore, remain, to this day, the most active molecules against uropathogenic *Enterobacteriaceae* [49].

The overall frequency of ESBL-producing *Enterobacteriaceae* among uropathogenic strains in this study was 6%. A similar prevalence (6.05%) was estimated in Morocco in 2014 [50]. Recently, the highest rate (56.2%) was reported in Benin among *Enterobacteriaceae* samples collected at the Cotonou National Center Hubert Koutoukou Maga university Hospital, Benin [51]. The differences observed from the study conducted in Benin may be due to the fact that their samples included various infections, whereas our sampling targeted only UTIs. In Europe, the resistance of *Enterobacteriaceae* to third-generation cephalosporins ranged from 6.2 to 30.8% among bacterial isolates in 2019 [52]. Muriuki et al. [53] reports a similar finding for uropathogenic *E. coli* in Kenya between 2015 and 2018. The production of ESBL is probably due to the often-empirical prescriptions of ß-lactams, particularly in ambulatory medicine, while awaiting the results of ECBU. The development of resistance to third-generation cephalosporins is a major cause of prolonged hospitalization of infected patients and limits treatment options agaimst the bacteria [54]. A 25% prevalence of ESBL production, therefore, creates significant therapeutic problems and will limit or reduce treatment options [55]. Our results on the production of ESBL by our strains of uropathogenic *Enterobacteriaceae* are, therefore, reassuring, but should be monitored.

Of the 103 *Enterobacteriaceae* isolates tested for biofilm production, 40% formed a biofilm on the microplate. These results are similar to other previous studies [56]. Biofilms provide a survival strategy for bacteria by positioning them to efficiently use available nutrients and prevent access to antimicrobial agents, antibodies and white blood cells [57]. It has also been found that biofilms harbor a large number of enzymes that inactivate antibiotics, such as beta-lactamases, and thus, create an island of antimicrobial resistance [58]. Biofilms, therefore, make our strains more virulent and multi-resistant. Recent studies have revealed that a reduction in oxygen tension in the bladder, combined with the presence of terminal electron receptors in the urine, facilitates the preferential expression of *E. coli* [59]. The expression of other factors, such as cytochrome bd quinol oxidase, promotes biofilm complexity and resistance to extracellular stressors by altering the abundance of extracellular matrix components [59]. This biofilm formation may be due to curli (functional amyloid) fibers, which constitute the main protein component of many biofilms of Gram-negative bacteria [60]. The presence of curli fibers in these biofilms provides a competitive advantage in mouse models of urinary tract infection by promoting adhesion to bladder epithelial cells [61]. This adhesion is further increased by the presence of phospho-ethanolamine cellulose produced simultaneously by UPECs [61].

Regarding genes encoding for virulence factors, in this study, the distribution of fimbriae was observed. *Enterobacteriaceae* harbor the *sfa* gene (25.15%). The *papG* gene is present in 9.71% and the *focG* gene in 8.74% of the *Enterobacteriaceae* strains. These results are lower than those obtained in Romania [62], in Mongolia [63] and in Egypt [64]. Fimbriae are required by the bacterium to promote the colonization of surfaces, which helps prevent urinary outflow and allows for infection by the bacterium, which may indicate their critical role in the production and progression of urinary infections. They also play an important role in the formation of biofilms [65].

In our study, uropathogenic *Enterobacteriaceae* secreted toxins such as α-hemolysin (*hly*) and cytotoxic necrosis factor 1 (cnf1) with respective presence percentages of 2.91% and 15.53%. These results are similar to studies conducted in Romania (13%) [61] and Mexico (15.4%) [66]. These toxins promote the exfoliation of bladder cells and cell lysis, which makes available the iron and nutrients necessary for bacterial growth [29]. Alpha-hemolysin has been associated with clinical severity in patients with UI and CNF1 with bladder inflammation [67].

Genes encoding siderophores were detected in our study. *Enterobacteriaceae* possessed genes coding for salmochelins *IronB* (22.33%), followed by genes coding for aerobactin *iucD* (21.36%) and, finally, those coding for yersiniabactin *fyuA* (5.82%). Siderophores such as toxins allow the bacterium to mobilize iron. They are essential virulence factors in most pathogenic Gram-negative bacteria [68].

In the present study, we also identified the presence of an autotransporter (*sat* gene) in 6.79% of the isolated strains. This rate is much lower than the 31.1% found in Guadeloupe [29]. Autotransporters can be self-secreted through the membrane of Gram-negative bacteria [69] and modify the structure of the host cell. Thanks to these capacities, they make the strains more virulent.

The *sfa* gene was the most detected among the virulence genes in the present study. This high prevalence could lead us to consider *sfa* as a candidate for a potential vaccine. The difference in the prevalence of virulence genes between our studies and different studies abroad may be due to differences in sample size and methodology. Virulence factors are the product of different genes, which can be detected using the PCR method [70,71]. However, due to a possible mutation in the corresponding gene, PCR may not detect the presence of the gene [31]. Therefore, although this phenomenon is rare, a negative PCR result does not necessarily equate to the absence of the corresponding gene [31].

In our study, biofilm production is associated with the virulence gene *fyuA* and gentamicin resistance. This observation has been made in various other studies. Thus, Stephenson and Brown [72] reported that biofilm production was significantly associated with resistance to fluoroquinolones [72]. In another study by Zamani et al. [73], it was found that biofilm production in UPEC was significantly associated with the *Fim* gene [73]. Our study also found that the presence of *PapG*, *iucD* and *Iron*B virulence genes decreases biofilm production. On the other hand, in a survey carried out in Uganda [17], biofilm production was not associated with any virulence gene or resistance to a particular antibiotic. Thus, further studies are required to better understand the relationship between virulence factors, antibiotic resistance and biofilm formation.

## 5. Conclusions

The bacteriological analyses that we carried out on urinary tract infections made it possible to identify microbial diversity in this study. Our results showed noticeable diversity in the distribution of *Enterobacteriaceae* species, namely strains of *Escherichia coli*, *Serratia marcescens*, *Klebsiella ornithinolytica*, *Serratia liquefaciens*, *Citrobacter freundii*, *Enterobacter intermidius* and *Klebsiella pneumoniae*. The highlighting of their resistance to antibiotics revealed the importance of the frequency of multi-resistant strains to various antibiotics that are specifically used for treatment. In addition, the majority of strains were biofilm-forming, and a small proportion were ESBL-producing. To measure the danger that the strains can represent, we noted the presence of genes encoding for virulence factors. Therefore, the habituation of physicians to requesting a cytobacteriological examination of urine with an antibiogram is essential for any patient presenting suspicious signs of urinary tract infection. It is also necessary to plan a good strategy for the supply and dispensing of antibiotics to avoid self-medication. Empirical treatment of urinary tract infections in our country should also be revised. Further research on the main uropathogenic genes detected that can be used for the manufacture of potential vaccines must be carried out in this area.

## Figures and Tables

**Figure 1 microorganisms-11-00213-f001:**
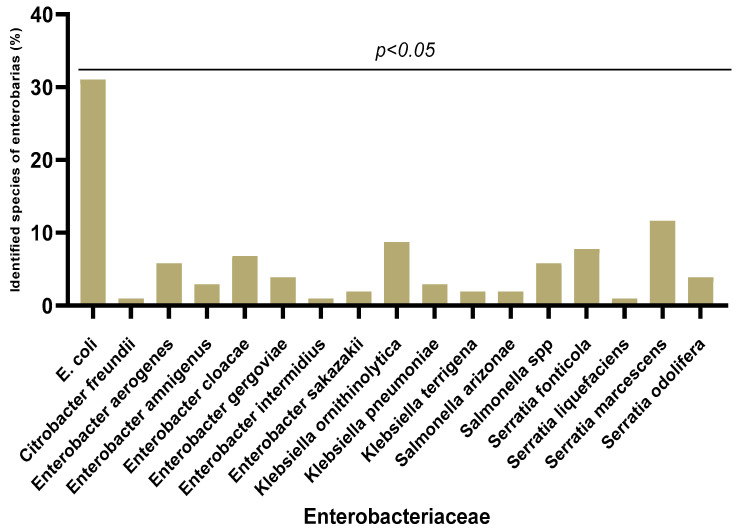
Different *Enterobacteriaceae* species isolated from urinary tract infections.

**Figure 2 microorganisms-11-00213-f002:**
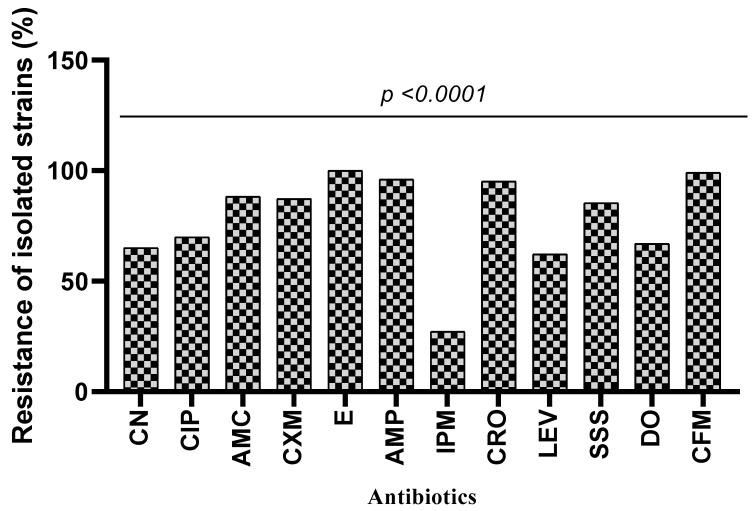
Resistance profile of isolated *Enterobacteriaceae* strains to antibiotics. CN: gentamicin, CIP: ciprofloxacin, AMC: amoxicillin/clavulanic acid, CXM: cefuroxime, E: erythromycin, AMP: ampicillin, IPM: imipenem, CRO: ceftriaxone, LEV: levofloxacin, SSS: sulfonamide, DO: doxycycline, CFM: cefixime.

**Figure 3 microorganisms-11-00213-f003:**
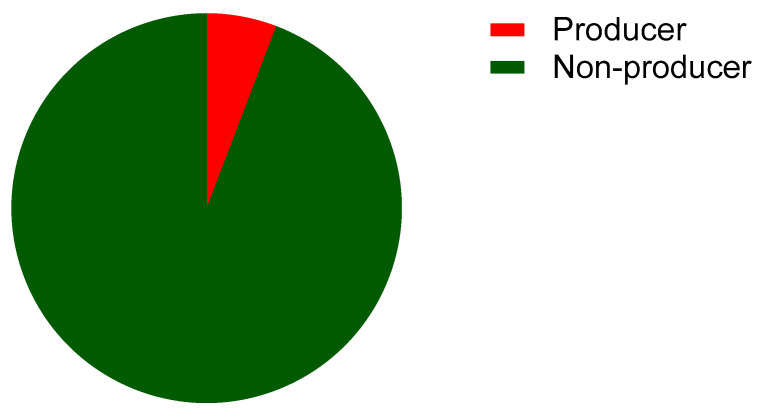
Proportion of ESBL producers among the enterobacterial strains isolated from urinary tract infections in Benin.

**Figure 4 microorganisms-11-00213-f004:**
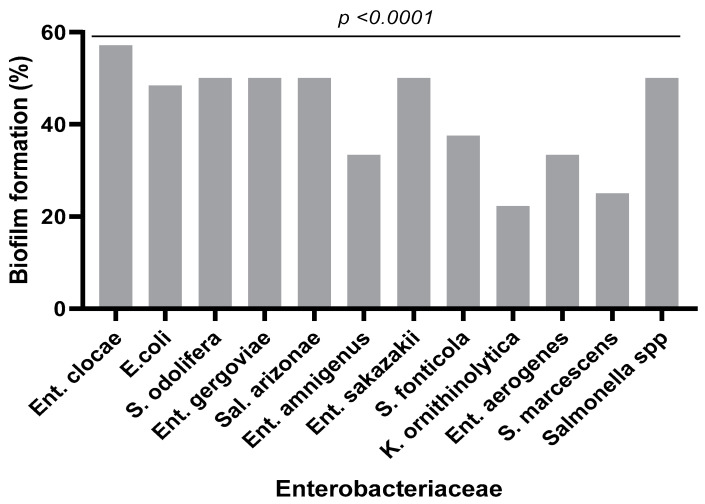
Biofilm production rate by bacterial species.

**Figure 5 microorganisms-11-00213-f005:**
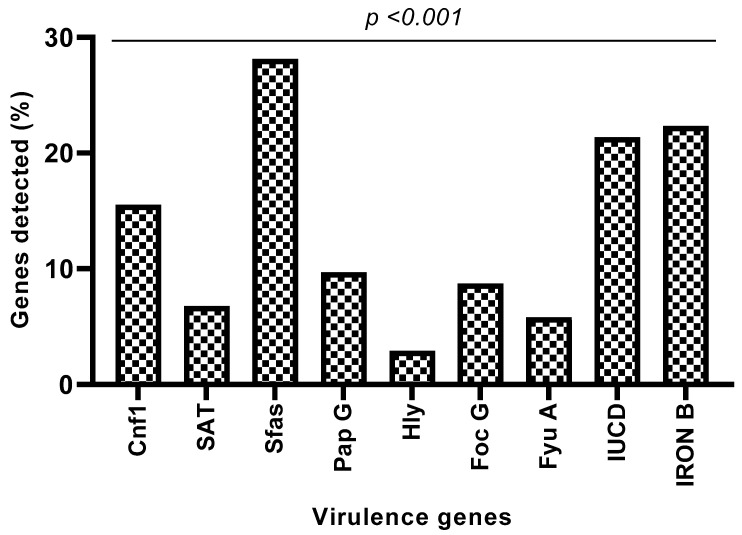
Proportion of genes sought in all strains. *Cnf1*: cytotoxic necrotizing factor type 1, *SAT*: secreted autotransporter toxin, *Sfas*: S-fimbriae adhesin, *Pap* G: adhesin P-fimbria, *Hly*: hemolysin RTX, *Foc* G: F1C fimbrial adhesin, *Fyu* A: yersiniabactin outer membrane receptor, *IUCD*: aerobactin biosynthesis receptor, *Iron*B: salmochelin outer membrane receptor.

**Figure 6 microorganisms-11-00213-f006:**
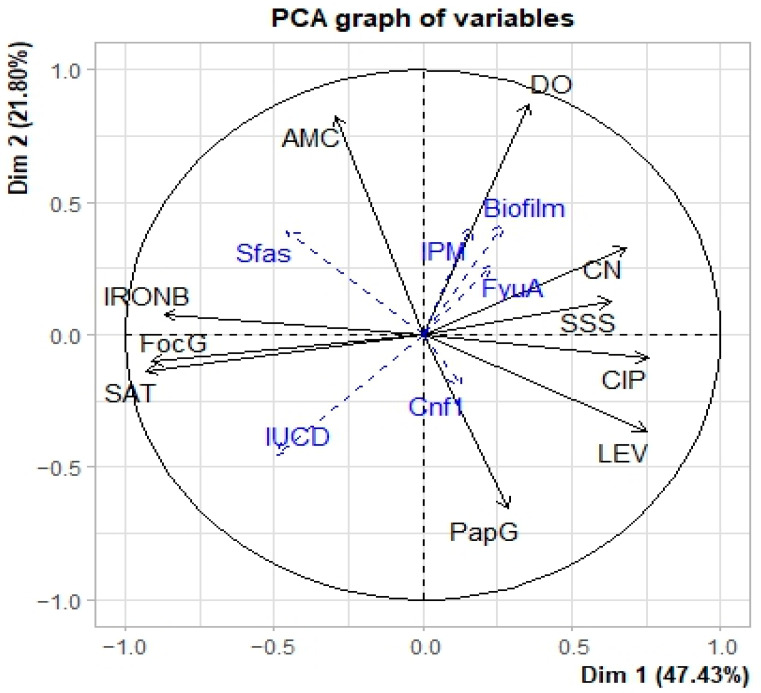
Circle of correlation of variables with the two axes.

**Figure 7 microorganisms-11-00213-f007:**
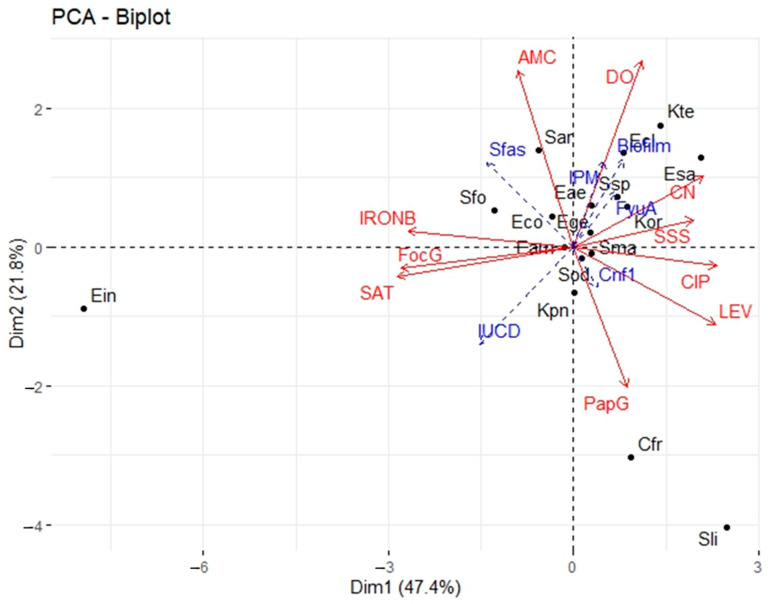
Scatterplot relating to the position of species with respect to axes 1 and 2. Legend: Cfr = *C. freundii*; Eco = *E. coli*; Eae = *E. aerogenes*; Eam = *E. amnigenus*; Ecl = *E. cloacae*; Ege = *E. gergoviae*; Ein = *E. intermidius;* Esa = *E. sakazakii*; Kor = *K. ornithinolytica*; Kpn = *K. pneumoniae*; Kte = *K. terrigena*; Sar = *S. arizonae*; Ssp = *S. spp*; Sfo = *S. fonticola*; Sli = *S. liquefaciens*; Sma = *S. marcescens*; Sod = *S. odolifera*.

**Table 1 microorganisms-11-00213-t001:** Sequences of the primers used for target genes.

Screened Gene	Primer	Primer Sequences (5′------->3′)	Expected Sizes (bp)
*cnf1*	Cnf1	5′-aagatggagtttcctatgcaggag-3′	498
	Cnf2	5′-cattcagagtcctgccctcattatt-3′	
*sat*	SAT F	5′-ggtattgatatctccggtgaac-3′	779
	SAT R	5′-atagccgcctgacatcagtaat-3′	
*papG II/III*	pF f	5′-ctgtaattacggaagtgatttctg-3′	1070
	pG r	5′-actatccggctccggataaaccat-3	
*iucD*	iucD f	5′-aaaactgacatcggatggc-3′	253
	iucD r	5′-gtatttgtggcaacgcagaa-3′	
*fyuA*	FyuA f’	5′-tgattaaccccgcgacgggaa-3′	880
	FyuA r’	5′-cgcagtaggcacgatgttgta-3	
*focG*	FocG f	5′-cagcacaggcagtggatacga-3′	360
	FocG r	5′-gaatgtcgcctgcccattgct-3′	
*sfaS*	SfaS f	5′-gtggatacgacgattactgtg-3′	240
	SfaS r	5′-ccgccagcattccctgtattc-3′	
*iroN*	IRON1	5′-tattcgtggtatggggccgga-3′	547
	IRON2	5′-gcccgcatagatattcccctg-3′	
*hlyA*	Hly f	5′-aacaasgataagcactgttctggct-3′	1177
	Hly r	5′-accatataagcggtcattcccrtca-3′	

*Cnf1*: cytotoxic necrotizing factor type 1, *SAT*: secreted autotransporter toxin, *Sfas*: S-fimbriae adhesin, *Pap* G: adhesin P-fimbria, *Hly*: hemolysin RTX, *Foc* G: F1C fimbrial adhesin, *Fyu* A: yersiniabactin outer membrane receptor, *IUCD*: aerobactin biosynthesis receptor, *Iron*B: salmochelin outer membrane receptor.

**Table 2 microorganisms-11-00213-t002:** Breakdown of patients according to sex and age.

Parameter	Variable	Percentage (%)
Sex	M	35.92
	F	64.08
Age	10–20 years	11.65
	21–30 years	22.23
	31–40 years	19.42
	41–50 years	6.8
	51–60 years	19.42
	61–70 years	16.5
	71–80 years	3.88

**Table 3 microorganisms-11-00213-t003:** Resistance to antibiotics by isolated species of *Enterobacteriaceae*.

Species	Antibiotics											
	CN	CIP	AMC	CXM	E	AMP	IPM	CRO	LEV	SSS	DO	CFM
*Escherichia coli*	61.29%	70.96%	96.77%	80.64%	100%	100%	22.58%	100%	58.06%	90.32%	58.06%	100%
*Citrobacter freundii*	0%	100%	0%	100%	100%	100%	0%	100%	100%	100%	0%	100%
*Enterobacter aerogenes*	66.66%	66.66%	100%	100%	100%	100%	33.33%	100%	66.66%	100%	66.66%	83.33%
*Enterobacter amnigenus*	66.66%	33.33%	66.66%	100%	100%	100%	66.66%	100%	33.33%	100%	33.33%	100%
*Enterobacter cloacae*	85.71%	71.42%	85.71%	85.71%	100%	100%	42.85%	85.71%	57.14%	100%	100%	100%
*Enterobacter gergoviae*	75%	25%	50%	75%	100%	100%	25%	100%	75%	100%	75%	100%
*Enterobacter intermidius*	0%	0%	100%	100%	100%	100%	0%	100%	0%	0%	0%	100%
*Enterobacter sakazakii*	100%	100%	100%	100%	100%	50%	50%	100%	100%	100%	100%	100%
*Klebsiella ornithinolytica*	66.66%	88.88%	88.88%	88.88%	100%	100%	33.33%	88.88%	66.66%	88.88%	77.77%	100%
*Klebsiella pneumoniae*	50%	100%	100%	50%	100%	50%	50%	100%	50%	0%	50%	100%
*Klebsiella ssp*	100%	100%	100%	100%	100%	100%	0%	100%	100%	100%	100%	100%
*Klebsiella terrigena*	100%	100%	100%	100%	100%	100%	0%	100%	50%	100%	100%	100%
*Salmonalla spp*	50%	66.66%	100%	83.33%	100%	100%	50%	100%	33.33%	100%	83.33%	100%
*Salmonella arizonae*	100%	100%	100%	100%	100%	100%	0%	100%	50%	50%	50%	100%
*Serratia fonticola*	37.50%	62.50%	75%	100%	100%	75%	12.50%	100%	37.50%	75%	75%	100%
*Serratia liquefaciens*	100%	100%	0	100%	100%	100%	0	100%	100%	100%	0	100%
*Serratia marcescens*	66.66%	66.66%	91.66%	91.66%	100%	100%	25%	75%	91.66%	75%	66.66%	100%
*Serratia odolifera*	75%	75%	75%	75%	100%	100%	0%	100%	75%	50%	50%	100%

CN: gentamicin, CIP: ciprofloxacin, AMC: amoxicillin/clavulanic acid, CXM: cefuroxime, E: erythromycin, AMP: ampicillin, IPM: imipenem, CRO: ceftriaxone, LEV: levofloxacin, SSS: sulfonamide, DO: doxycycline, CFM: cefixime.

**Table 4 microorganisms-11-00213-t004:** Proportion of genes recorded by species.

Species	Frequency of Virulence Genes (%)								
	*Cnf1*	*SAT*	*Sfas*	*Pap G*	*Hly*	*Foc G*	*Fyu A*	*IUCD*	*IronB*
*C. freundii*	100	0	0	0	0	0	0	0	0
*E. coli*	19.35	6.45	25.8	9.67	0	16.12	12.9	35.48	32.25
*En. aerogenes*	0	0	16.66	16.66	0	0	16.66	16.66	33.33
*En. amnigenus*	33.33	0	33.33	0	0	0	0	33.33	0
*En. cloacae*	0	0	28.57	0	0	14.28	0	14.28	0
*En. gergoviae*	0	0	0	0	0	0	0	0	25
*En. intermidius*	0	100	100	0	0	100	0	100	100
*En. sakazakii*	0	0	100	0	0	0	50	0	0
*K. ornithinolytica*	11.11	0	33.33	11.11	11.11	0	0	0	11.11
*K. pneumoniae*	0	0	0	50	50	0	0	0	0
*K. terrigena*	50	0	50	0	0	0	0	0	0
*S. arizonae*	50	0	50	0	0	0	0	50	50
*Salmonella spp*	0	0	16.66	0	16.66	0	0	16.66	0
*Serratia fonticola*	12.5	12.5	25	0	0	12.5	0	12.5	50
*S. liquefaciens*	0	0	0	100	0	0	0	100	0
*S. marcescens*	33.33	16.66	41.66	25	0	0	0	16.66	25
*S. odolifera*	0	0	0	0	0	25	0	0	0

*Cnf1*: cytotoxic necrotizing factor type 1, *SAT*: secreted autotransporter toxin, *Sfas*: S-fimbriae adhesin, *Pap* G: adhesin P-fimbria, *Hly*: hemolysin RTX, *Foc* G: F1C fimbrial adhesin, *Fyu* A: yersiniabactin outer membrane receptor, *IUCD*: aerobactin biosynthesis receptor, *Iron*B: salmochelin outer membrane receptor.

**Table 5 microorganisms-11-00213-t005:** Eigenvalues of the correlation matrix.

	Main Components			
Settings	Dim,1	Dim,2	Dim,3	Dim,4
Own value	4.74	2.18	1.13	0.66
Percentage of variance	47.43	2.,80	11.28	6.62
Cumulative percentage of variance	47.43	69.23	80.51	87.13

**Table 6 microorganisms-11-00213-t006:** Correlation between the starting variables and the principal components.

	Dim,1	Dim,2
IRON.B	−0.870	0.076
SAT	−0.928	−0.136
Foc.G	−0.911	−0.100
AMC	−0.292	0.828
DO	0.358	0.873
LEV	0.751	−0.365
CN	0.685	0.331
CIP	0.755	−0.090
SSS	0.632	0.128
Pap.G	0.285	−0.657

Foc G: F1C fimbrial adhesin, SAT: secreted autotransporter toxin, Pap G: adhesin P-fimbria, IronB: salmochelin outer membrane receptor, AMC: amoxicillin/clavulanic acid, DO: doxycycline, LEV: levofloxacin, CN: gentamicin, CIP: ciprofloxacin, SSS: sulfonamide.

## Data Availability

The data used to support the findings of this work are available from the corresponding author upon request.

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
