# Peer review of "Susceptibility and Virulence of Enterobacteriaceae Isolated from Urinary Tract Infections in Benin"

_microorganisms, 2023, doi:10.3390/microorganisms11010213_

Round 1

Reviewer 1 Report (Previous Reviewer 3)

Dear authors,

After the review process, I have several comments: the authors should insert the paper in the journal format; the tables/figures do not have statistical data in the data presentation - it is essential for good research; the discussion section should be improved, starting from the presentation of in vitro resistance in urinary infection in a lab catheterization model.

Best regards!

Author Response

Comments and Suggestions for Authors

After the review process, I have several comments: the authors should insert the paper in the journal format; the tables/figures do not have statistical data in the data presentation - it is essential for good research; the discussion section should be improved, starting from the presentation of in vitro resistance in urinary infection in a lab catheterization model.

Answer to the reviewers’ Comments

We would like to sincerely thank the reviewer for his comments.

The manuscript has been revised using the journal format. The statistical data are provided some times in the main manuscript (p-values) and sometimes in the tables (5 and 6) / figures (6 and 7) of the revised manuscript (See the submitted revised manuscript). The discussion section has been improved correcting the English and the its consistence.

Reviewer 2 Report (Previous Reviewer 2)

The authors have submitted the revised version. However, I did not find a clear version, and reviewing the manuscript in this format is difficult. Please submit the manuscript again with both track and without track changes. 

Author Response

Comments and Suggestions of the Reviewer 2

The authors have submitted the revised version. However, I did not find a clear version, and reviewing the manuscript in this format is difficult. Please submit the manuscript again with both track and without track changes.

Answer to the Reviewer 2’ comment

Thank you for this comment. For the revised manuscript, we use the journal template. Thus, the line number is added in the revised manuscript. (See the MS word submitted revised manuscript with track changes).

Reviewer 3 Report (Previous Reviewer 1)

To my knowledge, the article submitted for review offers only epidemiological results from a geographic area for which we have little information. The revision of the article has improved the initial article. I would accept the article submitted for publication in the current version. 

Author Response

Comments and Suggestions for Authors

To my knowledge, the article submitted for review offers only epidemiological results from a geographic area for which we have little information. The revision of the article has improved the initial article. I would accept the article submitted for publication in the current version.

Answer to the reviewer 3’ Comments and suggestions

We would like to sincerely thank the reviewer for this final decision.

Round 2

Reviewer 1 Report (Previous Reviewer 3)

Dear authors,

Several aspects should be  realized: not all figures have statistical data included; in all figures statistical data should have the same representation; the authors should read again my first round of comments to improve discussion section.

Best regards!

Author Response

See attached 

Reviewer 2 Report (Previous Reviewer 2)

The authors have satisfactorily all my concerns diligently. No future questions from me.

Author Response

We would like to think the reviewer for this positive final decision.

Best regards 

This manuscript is a resubmission of an earlier submission. The following is a list of the peer review reports and author responses from that submission.

Round 1

Reviewer 1 Report

The article shows a small series of patients from Benin, analyzing the etiology and antibiotic susceptibility of the enterobacteria causing the infections studied. The real interest of the article is to know the situation of bacterial epidemiology and antibiotic resistance in a country of which there is little clinical information, which can make the reader draw conclusions by comparing with the situation in his environment. It is certainly a study with a small number of patients, but I consider its publication to be of interest, since it comes from a geographical area for which we have very little clinical information. 

Reviewer 2 Report

The authors reported "Antimicrobial Susceptibility and Virulence of Enterobacteriaceae isolated from urinary tract infections in Benin". The authors study antibiotic susceptibility and virulence factors reported from UTI n Benin. First of all the authors need to follow the journal guidelines during submission. As there is no line number in the manuscript and it is difficult for the reviewers to mention the comments here. Second the authors isolated Enterobacteriaceae strains from UTI however, they only detect Antibiotic susceptibility and Virulence strains, while they missed detecting different resistance genes which are involved in UTI. Also the drugs used are not enough. I have highlighted and commented directly in the PDF as there is no Line numbers which is difficult to mention all the comments here. Please go through the PDF manuscript and revised all the comments and suggestions briefly. Also add line numbers after resubmission.

Reviewer 3 Report

Dear authors,

After the review process, I have several comments: you should clearly present the aim of the paper in the abstract and introduction and eliminate the general data that could be part of the introduction; you should use the journal format when you upload the revised version of the paper; no statistical data, for example, table 1 and figures ...; in the discussion, you should include novel findings related to plant extract to prevent and treat recurrent infections instead of the first paragraph of this section; you should include, as future perspectives of the paper, new methods for testing urinary infection by lab catheterization model.

Best regards!